∂ | Open Peer Review | Veterinary Microbiology | Research Article

# Microbiota-accessible fiber activates short-chain fatty acid and bile acid metabolism to improve intestinal mucus barrier in broiler chickens

Jiantao Yang,[1] Kailong Qin,[1] Yanpeng Sun,[1] Xiaojun Yang[1]

**ABSTRACT** Dietary polysaccharides are closely associated with gut microbiota and intestinal function homeostasis. This study aimed to investigate the mechanisms by which the β-glucan, arabinoxylan, and resistant starch selectively modulate gut microbiota and microbiota-derived metabolites to protect the intestinal mucus barrier in broiler chicken. In the present study, cecal microbiota samples from 21-day-old broilers were cultured *in vitro* with β-glucan, arabinoxylan, or resistant starch as the sole carbon source. We found that β-glucan, arabinoxylan and resistant starch alter community structure and selectively target the enrichment of *Bacteroides*, *Lactobacillus, Coprococcus*, *Butyricicoccus*, *Ruminococcus* and *Blautia*, respectively. Notably, supplementing fiber-deficient diets with arabinoxylan and resistant starch, but not with β-glucan, improved the intestinal mucus barrier by cecal microbiota enrichment and increase in the concentration of microbiota-derived short-chain fatty acids (SCFAs). In addition, we illustrated through bacterial cultures *in vitro* that the supplementation of arabinoxylan and resistant starch resulted in a change in bacterial biotransformation of secondary bile acids. Our findings provide insight into how arabinoxylan and resistant starch may selectively target gut microbiota and mediate the production of SCFAs and bile acid biotransformation to improve intestinal mucus barrier function.

**IMPORTANCE** The intestinal mucus barrier, located at the interface of the intestinal epithelium and the microbiota, is the first line of defense against pathogenic microorganisms and environmental antigens. Dietary polysaccharides, which act as microbiota-accessible fiber, play a key role in the regulation of intestinal microbial communities. However, the mechanism via which dietary fiber affects the intestinal mucus barrier through targeted regulation of the gut microbiota is not clear. This study provides fundamental evidence for the benefits of dietary fiber supplementation in broiler chickens through improvement in the intestinal mucus barrier by targeted regulation of the gut ecosystem. Our findings suggest that the microbiota-accessible fiber-gut microbiota-short-chain fatty acid/bile acid axis plays a key role in regulating intestinal function.

**KEYWORDS** intestinal mucus barrier, polysaccharide, microbiota, broiler, arabinoxylan and resistant starch

Chicken meat is popular as a low-fat, low-cholesterol, high-protein food (1). Its consumption has been increasing, and it is currently the second largest source of meat protein (2). As a result of crowded breeding conditions and various stressors, poultry are susceptible to diseases. In particular, the immune system of broilers has low immune activity in the early stages of development (3). Antibiotics can fight the invasion of pathogens and viruses and reduce inflammation by regulating the gut microbiome.

Address correspondence to Xiaojun Yang, yangxj@nwsuaf.edu.cn.

The authors declare no conflict of interest.

See the funding table on p. 16.

However, the use of antibiotics in the production of animals in many countries has been banned due to the development of antibiotic resistance (4). Therefore, improving the early health status of poultry is an important strategy from an economic perspective.

The intestinal mucus barrier, which is located at the interface of the intestinal epithelium and the microbiota, is important for preventing the entry of harmful substances, including microorganisms, foreign antigens, and toxins (5). Destruction of the intestinal mucus barrier is associated with bacterial translocation, systemic inflammatory response, celiac disease, and metabolic diseases (6–8). Mucins secreted by goblet cells provide the structural framework of the intestinal mucus barrier (9, 10). Mucins are a family of glycosylated proteins that contain glycoproteins, proteins, lipids, and water, which confer moisturizing and lubricant properties to mucus (10, 11). The intestinal mucus barrier is considered to contain crucial factors for physiological defense against mechanical, chemical, and biological attacks.

The gut microbiota is a complex and dynamic ecosystem that plays key roles in nutrient digestion, gut development, resistance to pathogens, and protection of the gut barrier (12, 13). Substantial research studies have demonstrated that the gut microbiota can influence the properties of the intestinal mucus barrier (14, 15). This effect is mediated via a wide range of small-molecule metabolites, such as short-chain fatty acids (SCFAs) and bile acids, secreted by gut microbes (12). Recent studies have shown that the production of SCFAs by gut microbes occurs through microbial fermentation of carbohydrates, which act as a source of energy for epithelial cells and provide protection to the intestinal mucus barrier (15, 16). In addition, the gut microbiota-derived bile acids serve as natural agonists for farnesoid X receptor and G protein-coupled receptor, and the bile acids chenodeoxycholic acid (CDCA) and deoxycholic acid (DCA) play a crucial role in maintaining homeostasis of the intestinal barrier (5, 17). These findings suggest that targeted regulation of the microbiota may be an effective strategy for maintenance of intestinal homeostasis.

The main source of microbiota-accessible fiber (MAF) in the diet is plants, as this type of fiber is the main component of the plant cell wall. It cannot be degraded by the host's endogenous carbohydrate enzymes but, instead, can be degraded and ingested by gut microbiota (18). In fact, a large proportion of the genome of microbes found in the gut are involved in the degradation and uptake of fiber. For example, *Bacteroides* are major members of the gut microbiota, devoting up to ~20% of their genomic content to the uptake and breakdown of polysaccharides (19). The scarce dietary fiber disturbs the fine-scale spatial architecture of gut microbiota and the community composition (20). Corn-soybean meal diets given to poultry contain abundant dietary fibers, such as β-glucan, arabinoxylan, and resistant starch, which are broadly defined as anti-nutrient factors (21). The vast majority of studies suggest that dietary fiber can alter the microbiota to improve intestinal mucosal barrier function (6, 13, 22, 23). Furthermore, specific components of fibers can selectively target microbes in the gut ecosystem to influence host health (24, 25). As β-glucan, arabinoxylan, and resistant starch are found in high concentrations in the dietary fiber of cereal grains and legumes, in the present study, we decided to investigate the effects of these three components on the intestinal mucus barrier via their effect on the gut microbiota of broiler chickens.

In this study, we hypothesized that β-glucan, arabinoxylan, and resistant starch improve intestinal mucus barrier function by selectively targeting gut microbiota and modulating the gut microbial metabolic pathways related to SCFAs and bile acids (BAs). Our results demonstrated that β-glucan, arabinoxylan, and resistant starch target gut microbiota under both *in vitro* conditions modeled in broiler chickens and *in vivo* conditions modeled in cecal microbial cultures. Further experiments confirmed the beneficial effects of arabinoxylan and resistant starch on the intestinal mucus barrier of broilers. Additionally, we found that arabinoxylan and resistant starch particularly affected bacterial SCFA production and BA transformation by altering gut microbiota composition and, thus, improved intestinal mucus barrier function. Overall, our data revealed the mechanisms via which dietary fiber targets gut microbiota and improves

intestinal mucus barrier function, thus providing new insights for nutrition interventions to improve the intestinal health of broiler chickens and increase productivity in the poultry industry.

## RESULTS

### *In vitro* fermentation characteristics of β-glucan, arabinoxylan, and resistant starch by broiler cecal microbiota

To investigate whether β-glucan, arabinoxylan, and resistant starch affected the cecal microbiota of broilers, *in vitro* fermentation experiments with β-glucan, arabinoxylan, or resistant starch as the sole carbon source were performed. Cecal microbiota exhibited efficient growth when arabinoxylan or resistant starch was the sole carbon source (Fig. 1A), and a significant increase in the concentration of lactic acid and SCFAs and resulting acidification of the medium were observed (Fig. 1B through I). Furthermore, β-glucan, arabinoxylan, and resistant starch selectively and significantly increased the contents of lactate, acetic acid, and isobutyric acid, respectively (Fig. 1B, C and E). Hence, β-glucan, arabinoxylan and resistant starch can be foraged as MAF for cecal microbiota of broilers

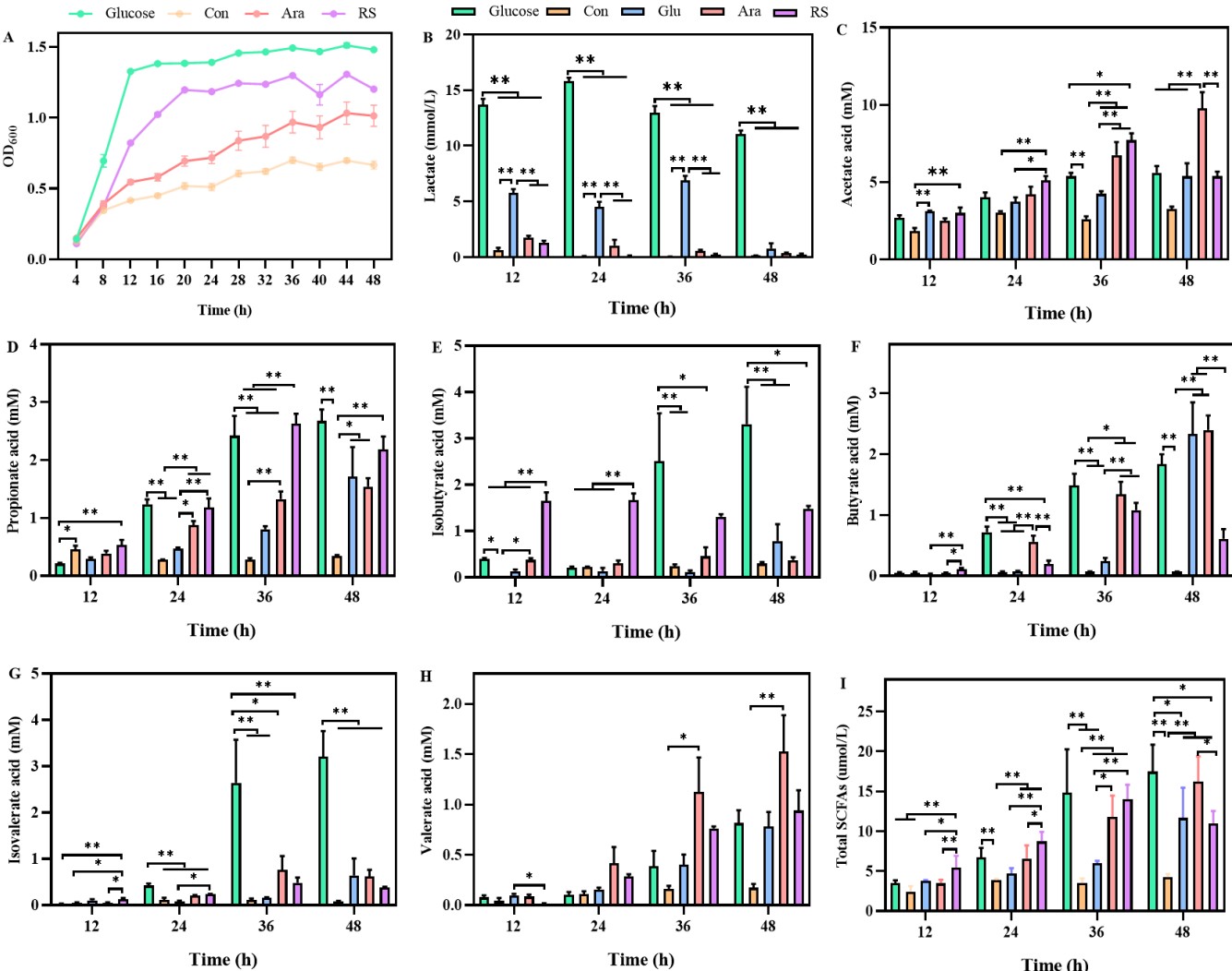

**FIG 1** *In vitro* fermentation characteristics of four polysaccharides. (A) Growth curves of cecal microbiota in yeast casitone fatty acids medium (YCFA medium) without carbon source (Con) or supplemented with 1% of either glucose (Glucose), arabinoxylan (Ara), or resistant starch (RS). Change in supernatant concentrations of (B) lactate and (C–I) SCFA during cecal microbiota growth on glucose, β-glucan (Glu), Ara, or RS. All data are expressed as mean ± SEM ($n = 5$). One-way analysis of variance was performed followed with *post hoc* Tukey's test. *$P < 0.05$, ** $P < 0.01$.

and exhibit exclusive metabolic characteristics, implying that may specifically primary target degraders.

## β-glucan, arabinoxylan, and resistant starch alter the broiler cecal microbiota *in vitro*

Microbiota fermentation samples collected throughout growth were characterized using 16S rRNA amplicon sequencing. We analyzed the microbiota structure of 60 samples based on Bray-Curtis distances. The overall community structure showed distinct differences between the glucose, β-glucan, arabinoxylan, and resistant starch groups, but samples within the same group showed no significant changes over time [permutational multivariate analyses of variance (PERMANOVA), $P < 0.05$, Fig. 2A].

At the phylum level, Firmicutes, Bacteroidetes, Proteobacteria, and Actinobacteria were the dominant phyla that accounted for over 95% of the phyla in all the samples (Fig. 2B). Compared to the control and glucose diets, the β-glucan and resistant starch diets caused a significant decrease in the relative abundance of Firmicutes and a significant increase in the relative abundance of Bacteroidetes species, which typically encode a large number of polysaccharide utilization loci (Fig. 2C). At the genus level, the fermentation enriched a high abundance of *Bacteroides*, *Enterococcus*, *Lactobacillus*, and *Alistipes* (Fig. 2D). Notably, β-glucan caused a significant increase in the relative abundance of

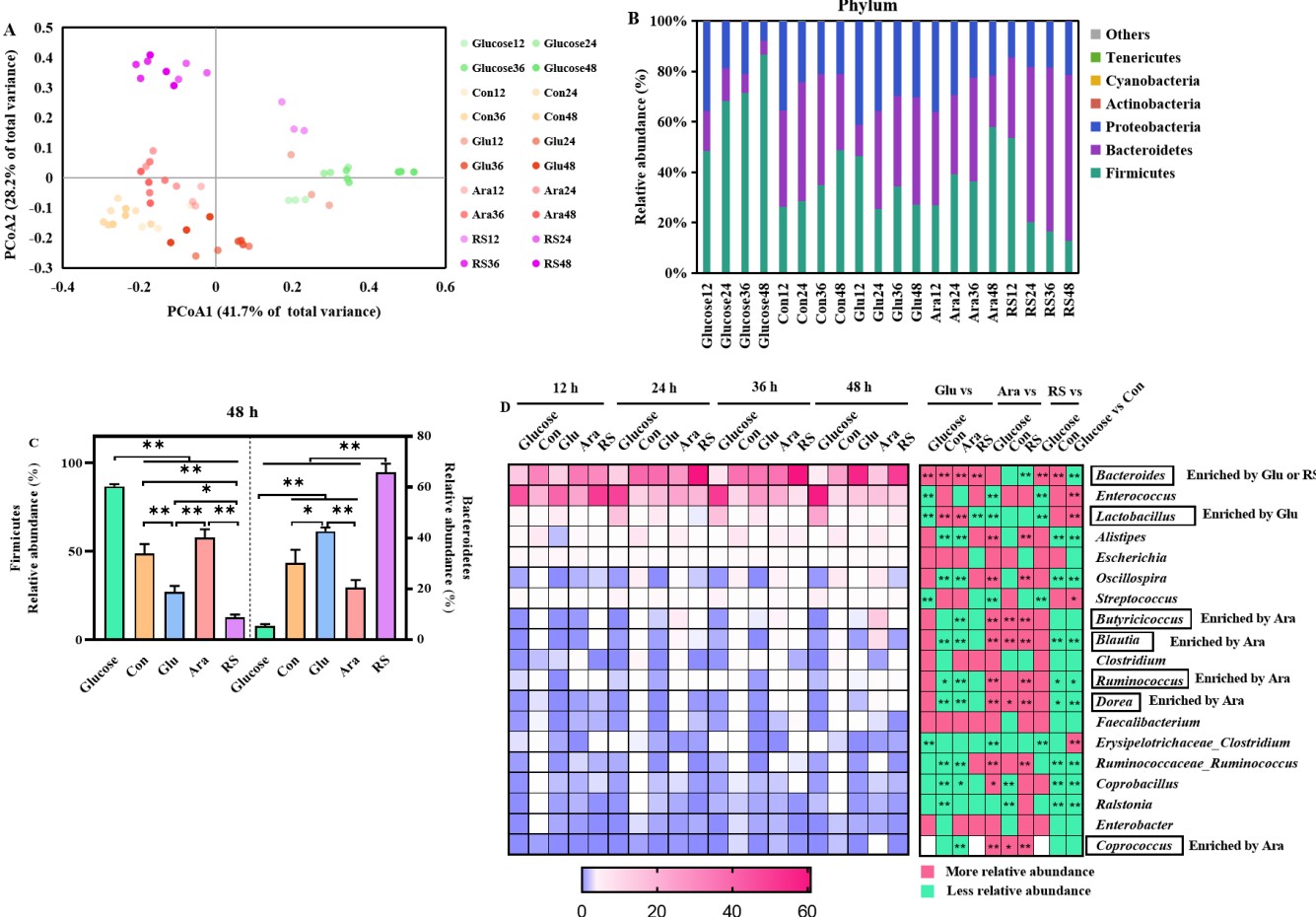

**FIG 2** Response of cecal microbiota to β-glucan, arabinoxylan, or resistant starch *in vitro*. (A) Overall cecum microbiota structure. Principal coordinate analysis (PCoA) was performed on the basis of the Bray-Curtis distance. (B) Relative abundance of bacteria at phylum. (C) The change of Firmicutes and Bacteroidetes. (D) The mean relative abundance of top 20 genera, the right side shows the difference analysis at 48 h. All data are expressed as mean ± SEM (*n* = 3). Significance between community structures was evaluated by PERMANOVA. Data were analyzed using one-way analysis of variance followed by Benjamini and Hochberg's *post hoc* test and false discovery rate correction for multiple testing. *$P < 0.05$, **$P < 0.01$.

*Bacteroides* and *Lactobacillus* at 48 h (Fig. 2D), and arabinoxylan caused a significant increase in the relative abundance of *Butyricicoccus*, *Ruminococcus*, and *Blautia* at 48 h (Fig. 2D). Furthermore, *Bacteroides* was mainly enriched in resistant starch supplementation treatment at 48 h (Fig. 2D). Together, the results obtained from the *in vitro* fermentation experiments illustrate the effects of β-glucan, arabinoxylan, and resistant starch on the community structure and microbiota composition of the gut in broiler chickens, as well as their specific targets.

## Effects of β-glucan, arabinoxylan, and resistant starch supplementation on growth performance

The effects of dietary β-glucan, arabinoxylan, and resistant starch supplementation on growth performance in broilers are shown in Fig. S1. Compared with the group fed a normal diet, the body weight (BW) of broilers on a dietary fiber-deprived diet was lower at 21 days, while the BW of the arabinoxylan and resistant starch supplementation groups was significantly higher (Fig. S1a). In addition, dietary fiber deprivation led to a significant decrease in average daily gain (ADG) and average daily feed intake (ADFI) over the experimental period of 1 to 21 days. In contrast, compared to the control group, the ADG levels were significantly higher in the arabinoxylan and resistant starch groups, and supplementation of resistant starch resulted in a significant increase in ADFI (Fig. S1b and c). However, no significant difference was found in feed conversion ratio (FCR) during the 1- to 21-day experimental period (Fig. S1d).

## Arabinoxylan and resistant starch improve intestinal mucus barrier in broilers

The mucins secreted by goblet cells maintain the integrity of the mucus barrier by continuous renewal (26). To determine whether β-glucan, arabinoxylan, and resistant starch affected the mucus layer, we fed broilers a standard maize-soybean meal diet (normal fiber level, NF), a dietary fiber-deprived diet (Con), or a dietary fiber-deprived diet supplemented with 3% β-glucan, arabinoxylan, or resistant starch for 10–21 days (Fig. 3A). Compared with the NF group, mucus layer injury was observed in the Con group (Fig. 3). Notably, arabinoxylan supplementation resulted in a significant reduction in intestinal permeability (Fig. 3F). Furthermore, an increase in the ileal thickness of the mucus layer was observed with arabinoxylan and resistant starch supplementation, but not with β-glucan supplementation (Fig. 3B and D). Consistent with these results, Alcian blue-periodic acid-Schiff (AB-PAS) staining showed that the amount of mucin was significantly greater in the ileum of broilers given an arabinoxylan- or resistant starch-supplemented diet (Fig. 3C and E). However, there were no significant differences in the expression of *Muc2, ZO1, Occludin*, or *Claudin* between broilers in the β-glucan, arabinoxylan, resistant starch, and Con groups (Fig. 3I and J). These results suggest that arabinoxylan and resistant starch supplementation, but not β-glucan supplementation, is beneficial to the intestinal mucus barrier.

## Alteration in cecal microbiota composition caused by β-glucan, arabinoxylan, and resistant starch supplementation

Diet is a key modulator of gut microbial architecture, and the exclusive metabolic niche formed by the polysaccharide-induced dynamics can influence the availability of nutrients to microbes (27, 28). Based on the results described in the previous subsection, we further hypothesized that the improvement in the intestinal mucus barrier was the result of gut microbiota regulation by arabinoxylan and resistant starch. To test this hypothesis, we used 16S rRNA sequencing to analyze the effect of β-glucan, arabinoxylan, and resistant starch on the cecum microbiota. Bray-Curtis distances revealed significantly different clustering of microbiota for each group (PERMANOVA, $P < 0.05$, Fig. 4A). Notably, significantly lower α-diversity, including Chao index and Shannon index, was observed with β-glucan and resistant starch supplementation (Fig. 4B).

We conducted further analysis to identify the bacterial species that were affected by β-glucan, arabinoxylan, and resistant starch. At the phyla level, the cecum harbored a

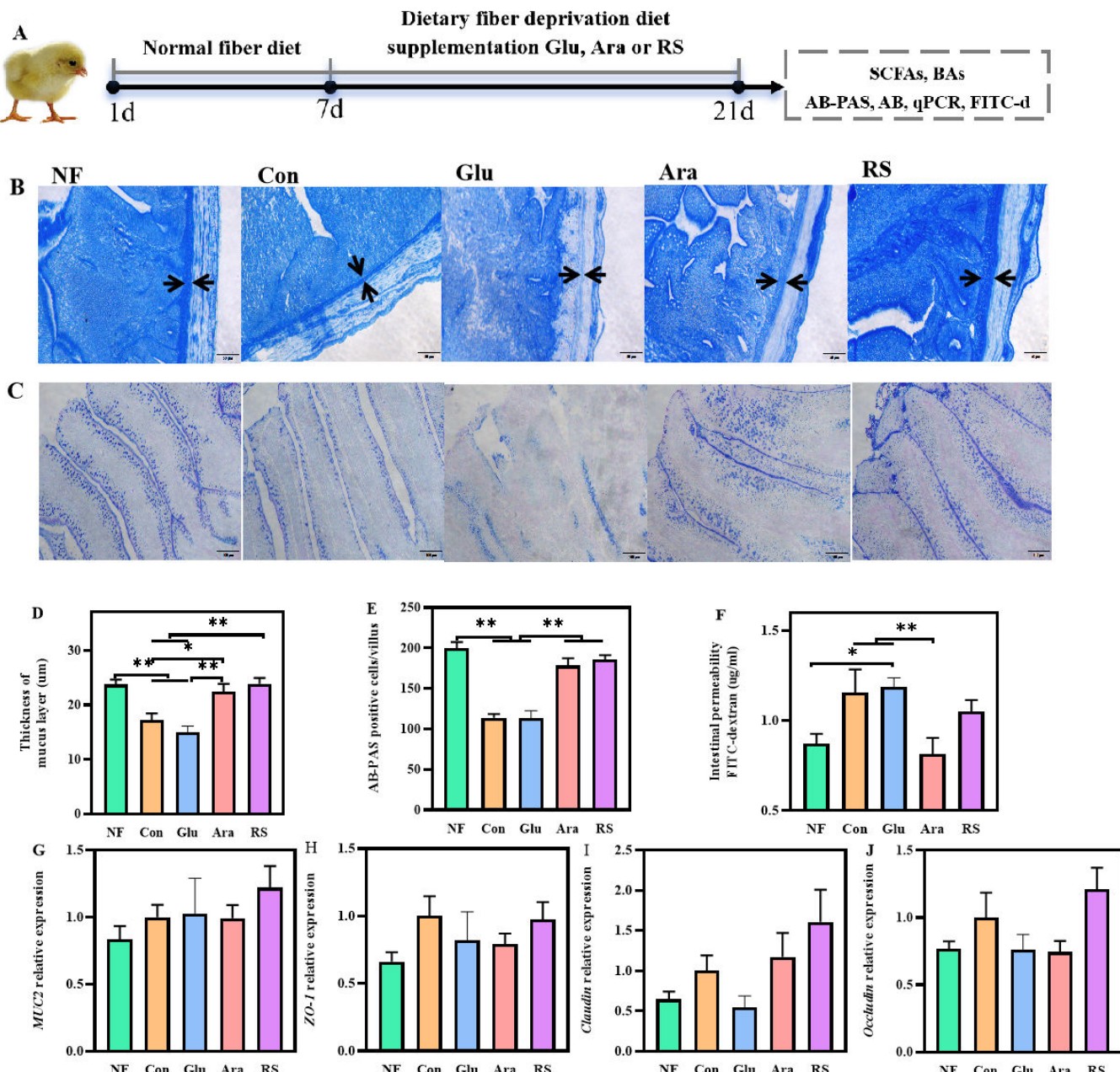

**FIG 3** Arabinoxylan and resistant starch improve intestinal mucus barrier. (A) Overview of experiment design. (B, D) Representative images of Alcian blue-stained ileal sections showing the thickness of mucus layer, blinded ileal mucus layer measurements shown on the right. (C, E) Representative images of Alcian blue-periodic acid-Schiff-stained ileal sections showing the number of mucin, the number of mucin shown on the right. (F) Intestinal permeability was measured as described in the Materials and Methods section. (G–J) The mRNA levels of barrier-related gene. All data are expressed as mean ± SEM ($n$ = 10). One-way analysis of variance was performed followed with *post hoc* Tukey's test. *$P$ < 0.05, **$P$ < 0.01.

high abundance of Firmicutes, Bacteroidetes, and Tenericutes (Fig. 4C). Interestingly, β-glucan and resistant starch resulted in significant enrichment of Bacteroidetes but a decrease in the relative abundance of Firmicutes (Fig. 4D). When compared with NF, the fiber-deficient diet (Con) caused a decrease in the relative abundance of Tenericutes (Fig. 4E). Broad changes in the inhabiting populations were found at the genus level, and the cecum was found to harbor a high abundance of *Oscillospira*, *Bacteroides*, *Lactobacillus*, and *Ruminococcaceae-Ruminococcus* (Fig. 4F). Consistent with the previously described fermentation experiments, β-glucan caused a significant increase in the relative abundance of *Lactobacillus* (Fig. 4F), and resistant starch caused significant enrichment of

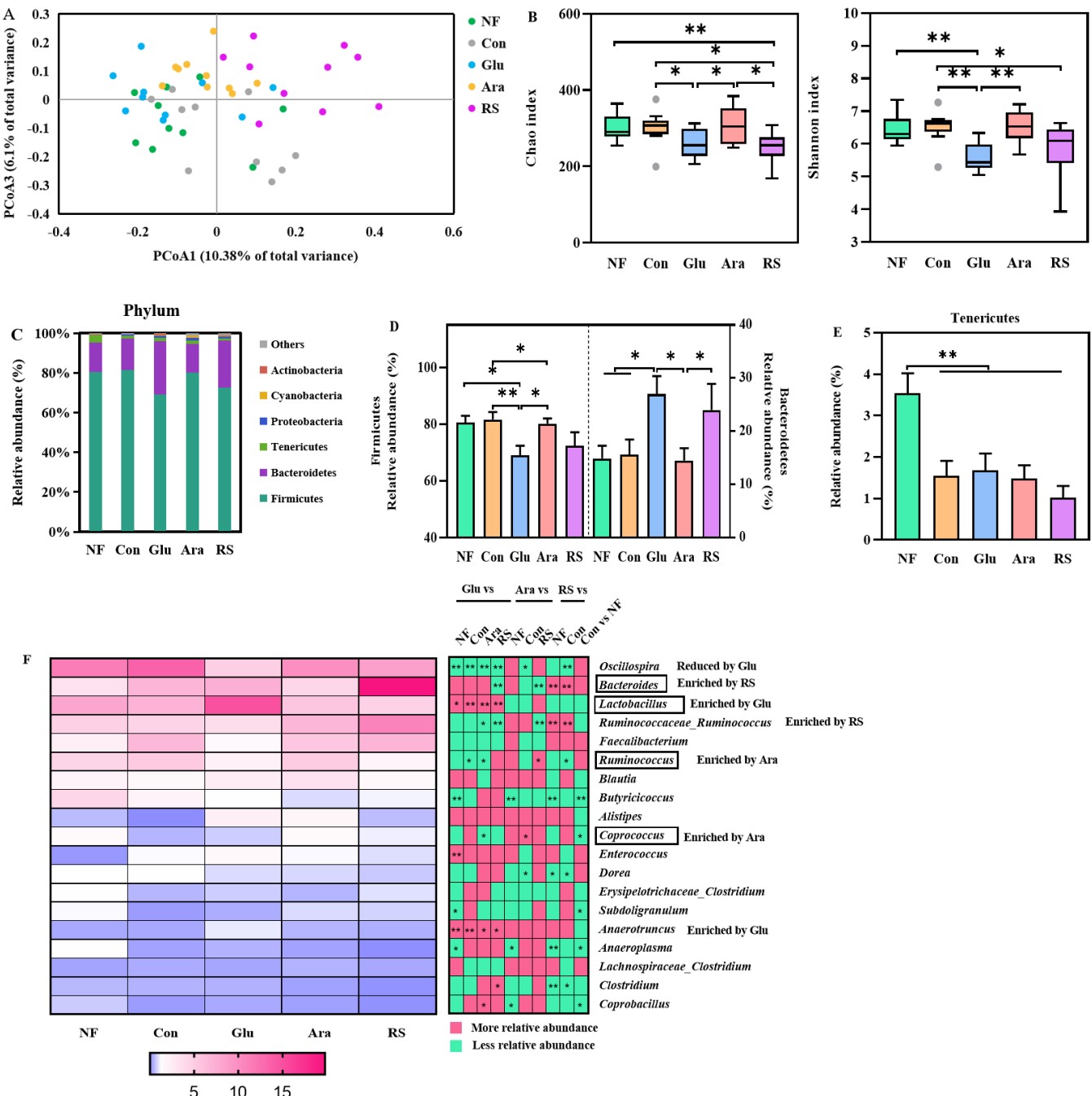

**FIG 4** Alter cecum microbiome profile following the supplementation of β-glucan, arabinoxylan, or resistant starch. (A) Overall cecum microbiota structure. Principal coordinate analysis (PCoA) was performed on the basis of the Bray-Curtis distance. (B) Overall community diversity of cecum microbiota. (C) Relative abundance of bacteria at phylum. (D) The change of Firmicutes and Bacteroidetes. (E) Relative abundance of Tenericutes. (F) The mean relative abundance of top 20 genera. All data are expressed as mean ± SEM (*n* = 10). Significance between community diversity was evaluated by Kruskal-Wallis test with *P*-value adjustment using false discovery rate (FDR) correction. Significance between community structures was evaluated by PERMANOVA. Data were analyzed using one-way analysis of variance followed by Benjamini and Hochberg's *post hoc* test and false discovery rate correction for multiple testing. *$P < 0.05$, **$P < 0.01$.

*Bacteroides* (Fig. 4F). However, the relative abundance of *Butyricicoccus* decreased significantly with the fiber-deficient diet (Fig. 4F). Moreover, arabinoxylan resulted in significant enrichment of *Coprococcus* but not *Blautia* or *Ruminococcus* (Fig. 4F). Bacterial interaction is a key factor in complex ecosystems, and therefore, arabinoxylan may have more varied effects on the gut microbiota. Thus, supplementation of β-glucan,

arabinoxylan, and resistant starch may have rapidly perturbed the gut ecosystem by selectively targeting certain microbes that play an important role in the intestinal mucus barrier.

## Increase in SCFA concentrations as a result of arabinoxylan and resistant starch supplementation

To investigate the effect of β-glucan, arabinoxylan, and resistant starch on microbiota-derived SCFAs, we evaluated the concentrations of SCFAs in the cecal lumen and feces of broiler chickens at 21 days (Fig. 5; Fig. S2). The concentration of isobutyric acid was significantly increased in both the cecum and feces in the resistant starch supplementation group (Fig. 5C; Fig. S2C). Moreover, compared with the NF group, the concentration of acetate acid was significantly increased in the cecum and feces of the arabinoxylan and resistant starch supplementation groups (Fig. 5A; Fig. S2A). In addition, the concentration of propionic acid was significantly decreased in the cecum but not in the feces following supplementation with arabinoxylan (Fig. 5B; Fig. S2B). However, butyric acid concentrations were similar across the groups (Fig. 5D; Fig. S2D). These findings, along with the findings of the fermentation experiments, indicate that supplementation with arabinoxylan and resistant starch may improve intestinal mucus barrier through an increase in the production of SCFAs by the microbiota.

## Alteration in bile acid composition caused by arabinoxylan and resistant starch supplementation

In this set of experiments, we tested the hypothesis that the effects of β-glucan, arabinoxylan, and resistant starch on the intestinal mucus barrier are a result of their effect on bile acid metabolism pathways of the gut microbiota. To prove this, we analyzed the concentrations of bile acids in the serum and liver of boiler chickens

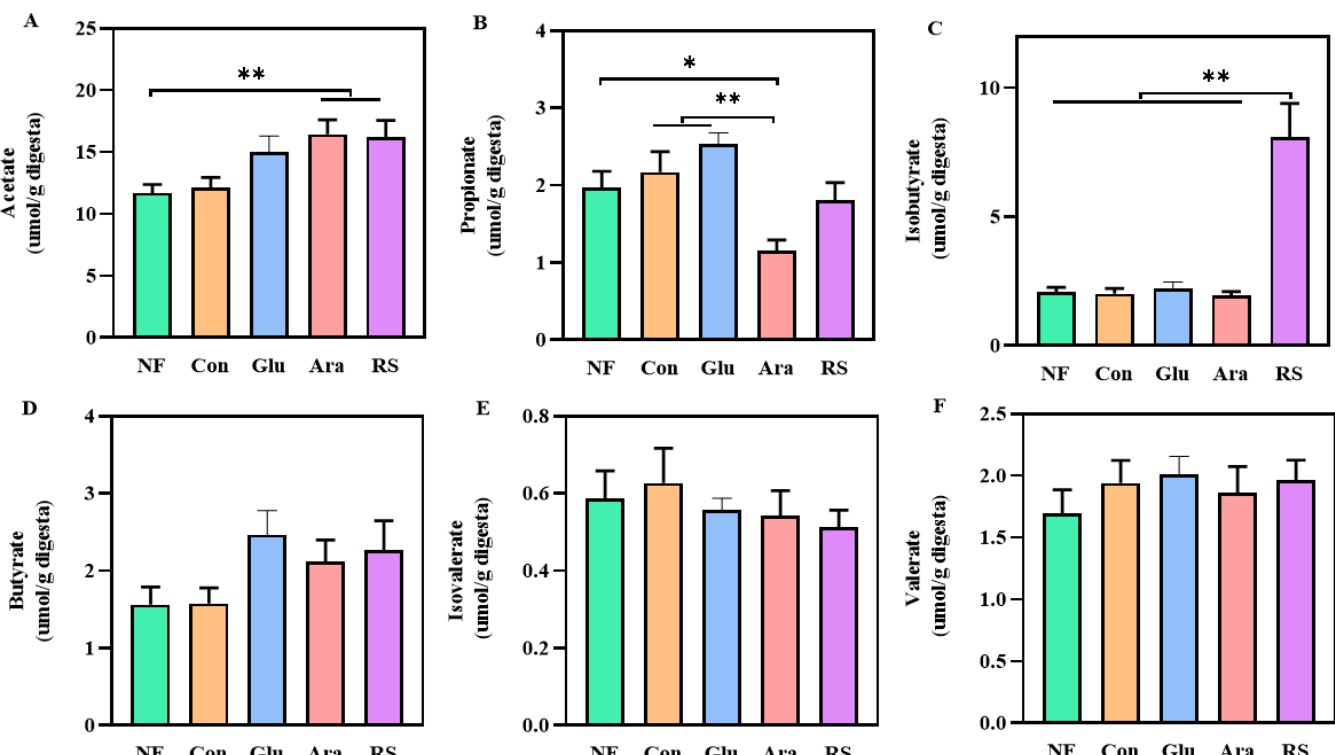

FIG 5 Altered gut microbiota profile by the supplementation of β-glucan, arabinoxylan, or resistant starch selectively promotes the production of SCFA. Changes in cecal concentrations of (A) acetic acid, (B) propionic acid, (C) isobutyric acid, (D) butyric acid, (E) isovaleric acid, and (F) valeric acid. All data are expressed as mean ± SEM ($n = 10$). One-way analysis of variance was performed followed with *post hoc* Tukey's test. *$P < 0.05$, **$P < 0.01$.

from all the groups by liquid chromatography-mass spectrometry (LC-MS). Compared with the Con group, the serum concentrations of cholic acid (CA), taurocholic acid (TCA), glycocholic acid (GCA), DCA, taurodeoxycholic acid (TDCA), and taurolithocholic acid (TLCA) were significantly increased, but the serum concentration of lithocholate acid (LCA) was significantly decreased, in the arabinoxylan group (Fig. 6). Interestingly, compared with the NF group, the concentration of LCA was significantly higher in the Con group (Fig. 6E). Arabinoxylan supplementation also caused a significant increase in the concentrations of TCA, GCA, TDCA, and TLCA in the liver (Fig. S3). Supplementation with β-glucan, arabinoxylan, and resistant starch caused a significant reduction in the concentrations of LCA in the liver compared with the NF and Con groups (Fig. S3E). These results suggest that the protective effects of arabinoxylan and resistant starch on the intestinal mucus barrier may involve the biotransformation of bile acids.

## Arabinoxylan and resistant starch supplementation impact bacterial metabolisms involved in secondary bile acid biotransformation

With strong bile salt hydrolase activity, gut microbiota have an important role in the biotransformation of bile acids (29). Accordingly, to confirm whether the alteration of microbiota mediated by β-glucan, arabinoxylan, and resistant starch supplementation directly impacts the productions of bile acids, we cultured cecal microbiota from broilers in the NF, Con, β-glucan, arabinoxylan, and resistant starch groups in brain heart infusion (BHI) medium containing bile acid of the gallbladder from Con broilers. Partial least squares discrimination analysis showed that bile acid production in the Con and NF

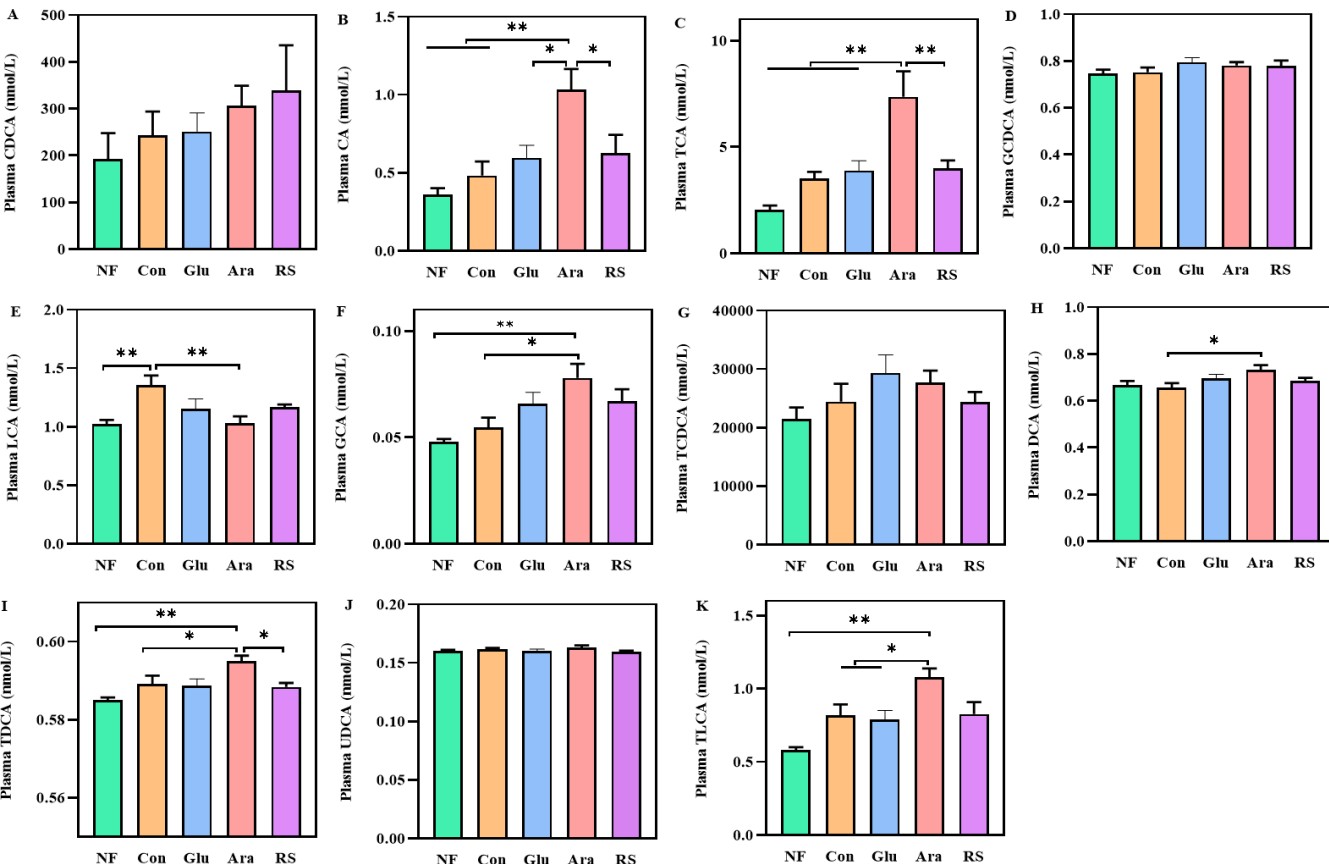

**FIG 6** Altered gut microbiota profile by the supplementation of arabinoxylan or resistant starch selectively increased the concentrations of bile acids. Changes in plasma concentrations of (A) CDCA, (B) CA, (C) TCA, (D) glycochenodeoxycholic acid (GCDCA), (E) LCA, (F) GCA, (G) taurochenodeoxycholic acid (TCDCA), (H) DCA, (I) TDCA, (J) ursodeoxycholic acid (UDCA), and (K) TLCA. All data are expressed as mean ± SEM ($n = 10$). One-way analysis of variance was performed followed with *post hoc* Tukey's test. *$P < 0.05$, **$P < 0.01$.

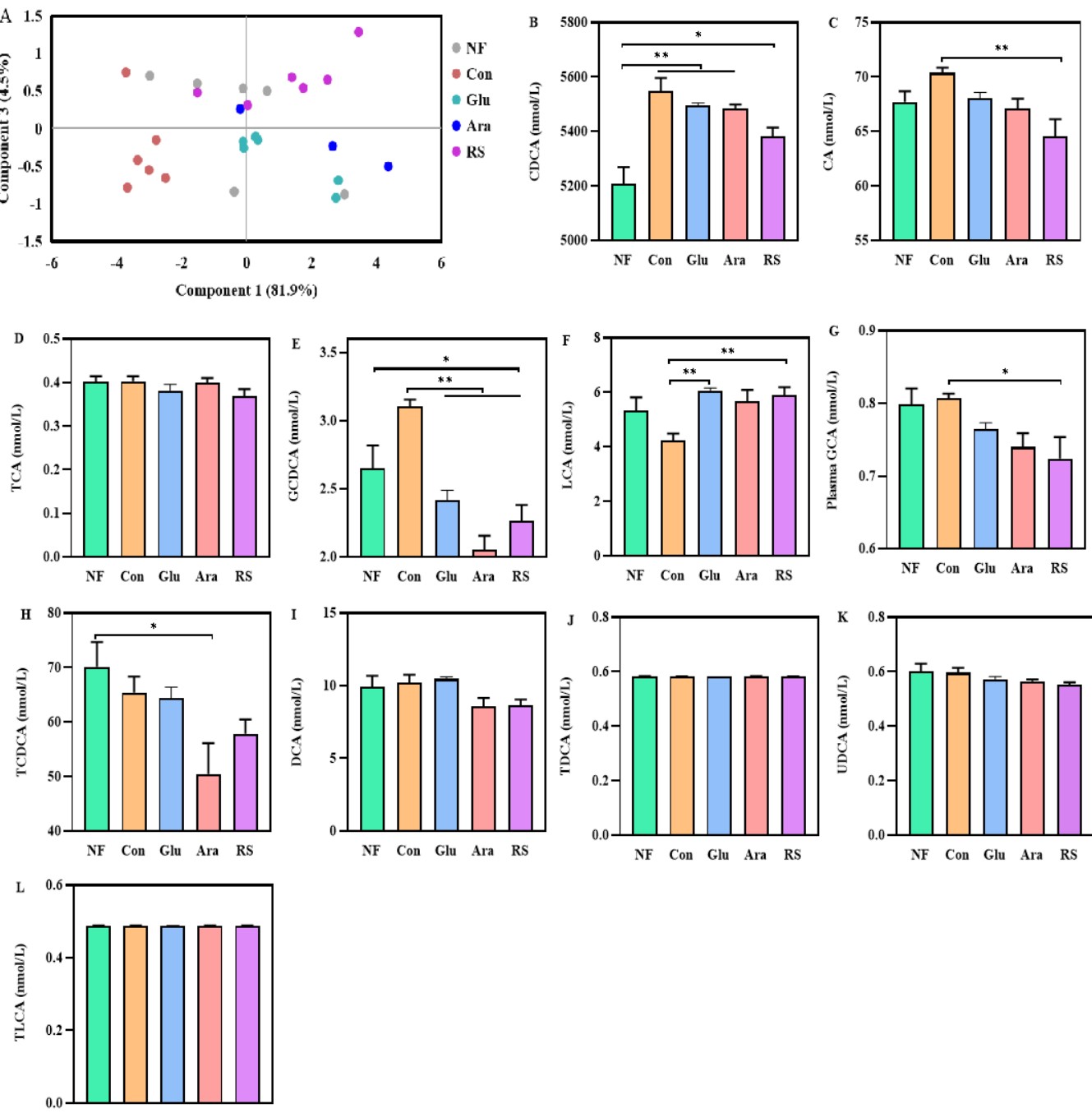

**FIG 7** Arabinoxylan or resistant starch supplementation altered the capacity of gut bacteria to produce bile acids *in vivo*. (A) Plots of the partial least squares discriminant analysis (PLS-DA) for the bile acids of supernatant. Changes in supernatant concentrations of (B) CDCA, (C) CA, (D) TCA, (E) GCDCA, (F) LCA, (G) GCA, (H) TCDCA, (I) DCA, (J) TDCA, (K) UDCA, and (L) TLCA. All data are expressed as mean ± SEM (*n* = 6). One-way analysis of variance was performed followed with *post hoc* Tukey's test. *$P < 0.05$, **$P < 0.01$.

groups was distinctly different from that in the β-glucan, arabinoxylan, and resistant starch groups (Fig. 7A). Specifically, the concentrations of CA and GCA were significantly lower in the resistant starch group than in the Con group (Fig. 7C and G). In contrast, the level of GCDCA was lower in the β-glucan, arabinoxylan, and resistant starch groups than in the Con group (Fig. 7E). Taken together, these findings indicate that bile acid production was metabolically regulated by β-glucan, arabinoxylan, and resistant starch.

## DISCUSSION

Dietary fiber in the form of MAF creates an exclusive metabolic niche for gut microbiota that presents an irreplaceable strategy for improving intestinal mucus barrier (30, 31). In the present study, we found that dietary supplementation of arabinoxylan and resistant starch protected the intestinal mucus barrier. Moreover, for the first time, we identified *Lactobacillus, Coprococcus,* and *Bacteroides* as targets of β-glucan, arabinoxylan, and resistant starch that were selectively enriched, through *in vitro* fermentation experiments and *in vivo* experiments on broilers fed various diets. In addition, the mechanism underlying the effects of these fibers on the intestinal mucus barrier was elucidated: arabinoxylan and resistant starch promoted the production of SCFAs and the biotransformation of bile acids by the gut microbiota. These results reinforce the notion that the MAF-gut microbiota-SCFA/bile acid axis has tremendous potential for protecting the intestinal mucus barrier.

The intestinal mucus barrier is the first line of defense against the migration of antigens and pathogens into the host (32). A growing body of evidence has suggested that dietary fiber deprivation can lead to intestinal barrier dysfunction (6). Polysaccharides have shown potent benefits in improving intestinal function because they selectively target gut microbes (33, 34). In line with these findings, intestinal morphological analysis in the present study showed that supplementation of arabinoxylan and resistant starch improved intestinal mucus barrier dysfunction caused by dietary fiber deprivation, although this effect was not observed with β-glucan supplementation. Improved intestinal permeability and goblet cell function are essential for fortification and protection of the intestinal mucus barrier (35). Accordingly, in our study, arabinoxylan and resistant starch supplementation were found to effectively increase the number of goblet cells and the thickness of the mucus layer. It is well recognized that the intestinal mucus barrier is a congenital barrier for maintaining the balance of the gut ecosystem (36). Thus, the findings collectively demonstrate that increase in intestinal permeability and the population of goblet cells induced by arabinoxylan and resistant starch contributed to the improvement of the intestinal mucus barrier.

The composition and function of the gut microbiota and the microbial metabolites produced have extensive physiological functions, including intestinal barrier protection, immune system regulation, and nutrient absorption (37, 38). Generally, the host genome lacks the enzymes required for the degradation of polysaccharides (18, 39). The main finding of this study is that arabinoxylan and resistant starch selectively targeted the intestinal microbiota to improve the intestinal mucus barrier. *In vitro* fermentation of cecal microbiota with β-glucan, arabinoxylan, and resistant starch as the only carbon sources showed that all three fibers altered the structure and composition of the intestinal microbial community. The degradation and uptake of glycan requires diverse carbohydrate-active enzymes, and *Bacteroides* are known to possess broad polysaccharide utilization loci with exceptional specificity for glycan (40). Interestingly, in the present study, supplementation of resistant starch significantly enriched the relative abundance of *Bacteroides* in the cecum. The size of ligands affects the bacterial capture of xylan. In addition to *Bacteroides*, some species of Firmicutes such as *Roseburia* species encode an impressive repertoire of carbohydrate-active enzymes and also degrade xylans. Xylan-degrading commensal *Bacteroides* species target larger ligands, but the transport protein of *Roseburia* displays a preference for oligomers (41). Further analysis in this study showed that *Coprococcus* was the dominant genus enriched in the arabinoxylan group, and *Coprococcus* species are fiber-fermenting SCFA producers (42). Notably, although β-glucan had no beneficial effect on the intestinal mucus barrier, it resulted in significant enrichment of *Bacteroides* and *Lactobacillus*, as reported previously by Tamura et al. too (43). Collectively, these findings indicate that arabinoxylan and resistant starch supplementation significantly altered the gut microbiota and also played a role in maintaining intestinal mucus barrier homeostasis.

In the gut ecosystem, SCFAs are the key bridge for maintaining the symbiotic relationship between the gut microbiota and the host; accordingly, promoting the

production of SCFAs from carbohydrate fermentation can enhance the function of the intestinal mucus barrier (44). For example, butyrate not only acts as an important energy substrate for goblet cells for MUC2 production, but also protects the integrity of the intestinal barrier (45). Previous publications have reported that *Coprococcus*, *Butyricicoccus*, *Ruminococcus,* and *Bacteroides* are important SCFA producers with excellent ability to produce SCFAs (46). In this study, we found that the content of SCFAs was selectively increased by arabinoxylan and resistant starch supplementation; thus, arabinoxylan and resistant starch might exert their protective effects on the intestinal mucus barrier by promoting the production of SCFAs by the gut microbiota.

The conversion of host-derived primary bile acids into secondary bile acids by gut microbiota is another key mechanism for improving intestinal barrier function (47). With strong bile salt hydrolase activity, *Coprococcus*, *Butyricicoccus*, *Ruminococcus,* and *Bacteroides* play an essential role in secondary bile acid biotransformation (48). This is similar to our finding that supplementation of arabinoxylan significantly increased the concentrations of TCA, GCA, TDCA, and TLCA in the liver and plasma. Accordingly, it has been reported that dietary fiber targets gut microbiota that mediate the secondary metabolism of bile acids (30, 49). Here, the dietary fiber-mediated cecal microbiota samples of the broilers were cultured in a medium containing gallbladder bile acids, and the results showed that supplementation of arabinoxylan and resistant starch impacted bacterial metabolic pathways related to secondary bile acid biotransformation. It has been reported that bile acids act as signaling molecules to maintain intestinal mucus barrier homeostasis (50, 51). Therefore, these findings collectively demonstrate that the protective effect of arabinoxylan and resistant starch in the intestinal mucus barrier was mediated by bile acid transformation.

In summary, the present findings imply that supplementation of arabinoxylan and resistant starch, but not β-glucan, can maintain intestinal mucus barrier function by selectively targeting *Coprococcus*, *Butyricicoccus*, *Ruminococcus*, and *Bacteroides* in the cecum, thereby resulting in an increase in the concentration of SCFAs and alteration in the composition of bile acids (Fig. 8). Our results provide fundamental evidence for

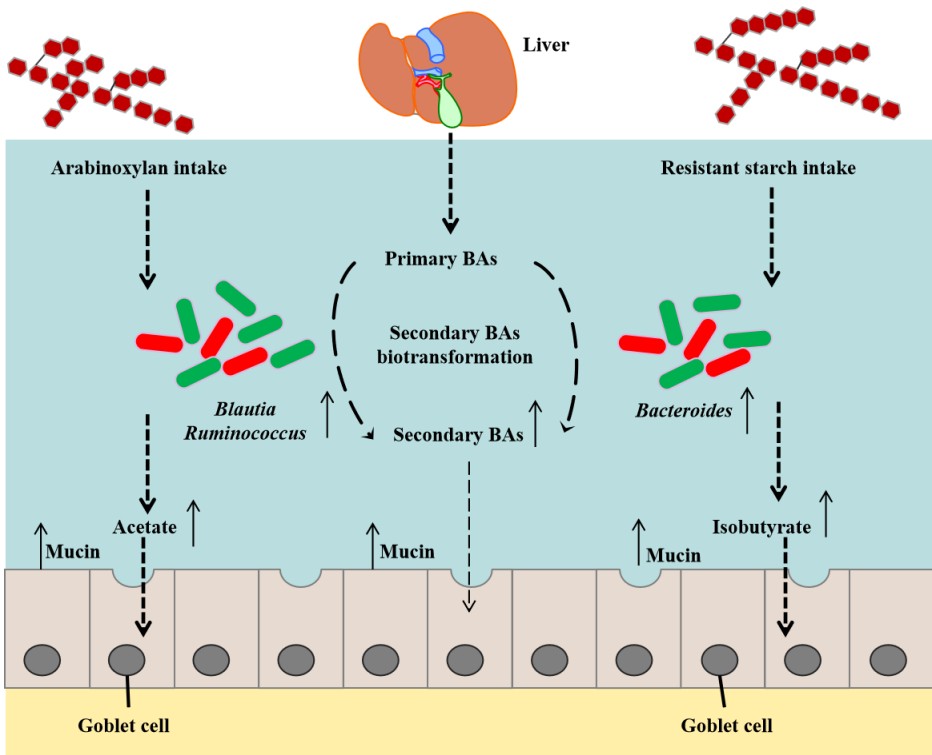

**FIG 8** Mechanism of arabinoxylan and resistant starch improves intestinal mucus barrier in a broiler model.

the role of dietary fiber in improving the intestinal mucus barrier through targeted regulation of the gut ecosystem. Thus, the MAF-gut microbiota-SCFA/bile acid axis may play a key role in the manipulation of intestinal function.

## MATERIALS AND METHODS

### *In vitro* cecal fermentation

Fresh cecal content were obtained from five healthy male broilers at 21 days and immediately transferred into an anaerobic chamber. Each cecal content (1 g) was diluted in 10 mL of a sterile Ringer working buffer [9 g/L of sodium chloride, 0.4 g/L of potassium chloride, 0.25 g/L of calcium dehydrate and 0.05% (wt/vol) L-cysteine hydrochloride] and vortexed until dispersed. Slurries were filtered through three layers of cheesecloth to remove visible particles. After centrifugation at 2,000 *g* at 4°C for 5 min, bacteria-enriched supernatants were collected. The supernatants (1 mL) immediately transferred into YCFA medium (10 mL) containing 1% (wt/vol) β-glucan (Glu) or arabinoxylan (Ara) or resistant starch (RS) or Glucose, and the medium without any carbon source served as negative control (Con). The mediums were incubated under anaerobic conditions at 37°C. Micobiota specimens were collected at 12, 24, 36, and 48 h for lactic acid, SCFA, and 16S rDNA analysis.

### Experimental design, housing, and diets

In the study, a total of 200 1-day-old Arbor Acres broilers were randomly divided into five groups. Each group contained 10 replicates with four birds per replicate. As shown in Fig. 3A, all broilers were fed a standard maize-soybean meal diet for the first weeks, after the diets were arranged as follows for 8–21 days: (1) standard maize-soybean meal diet (NF), (2) dietary fiber deprivation diet (Con), (3) dietary fiber deprivation diet with β-glucan (Glu), (4) dietary fiber deprivation diet with arabinoxylan (Ara), (5) dietary fiber deprivation diet with resistant starch (RS). β-glucan, arabinoxylan, and resistant starch were extracted from oat, maize, and maize, respectively. The ingredients and nutrient levels of the basal diets are shown in Table S1. All birds were fed in double-layer wired battery cages with *ad libitum* access to water and mash feed at the Experimental Teaching Center of Animal Science in the Northwest A&F University. Animals were housed in an environmentally controlled room with temperatures starting at 35°C and then decreased by 2°C every 7 days. The relative humidity was maintained at 55% to 65% for the first 2 weeks, then 45% to 55% for the third weeks. The lighting schedule was 23 h with 30 to 50 lx for the first week, and then reduced by 2 h per week with 25 lx. The study lasted 21 days. At 1 and 21 days, feed intake and BW were recorded for each replicate, then the ADFI, ADG, and FCR were calculated. At the end of the experiment, one male bird was selected from each replicate and serum samples were collected by centrifuging venous blood at 2,000 × *g* for 15 min at 4°C, and then euthanized by $CO_2$ asphyxiation, followed by cervical dislocation. Approximately 0.5 g of the ileal and cecal contents as well as 0.3 g of the feces were collected from each bird. The ileum was removed, the contents were removed with a cold phosphate buffered saline (PBS) wash, then the intestines were opened lengthwise to collect the mucosa. The 0.5 g of the liver was collected. All samples were flash frozen in liquid nitrogen and stored at −80°C until further processing.

### Lactic acid analysis

The concentration of lactic acid in fermentation supernatant was determined by using the reagent kit (Nanjing Jiancheng Technology Co., Ltd., Nangjing, China). The concentration of lactic acid was measured according to the manufacturer's instructions.

## SCFA profiling

The concentration of SCFAs in cecal contents and feces was determined by gas chromatography-mass spectrometry. Briefly, 0.3 g cecal contents and feces were homogenized with 1 mL of cold normal saline and centrifuged at 4°C at 13,000 $\times$ $g$ for 10 min. The supernatant was obtained and mixed with 0.4 mL metaphosphoric acid for deproteinization. After 4 h quiescence at 4°C, the mixture was centrifuged at 4°C at 13,000 $\times$ $g$ for 10 min. The supernatant was obtained and mixed with 0.2 mL crotonic acid. The mixture was filtered through a 0.45 µm nylon filter. Finally, the samples were measured by GC on an Agilent 6890 (Agilent Technologies, CA, USA) equipped with GC ChemStation software. Detailed methods were set according to reported methods (52).

## Intestinal permeability analysis measured by fluorescein isothiocyanate (FITC)-dextran assay

Intestinal permeability was assessed by the FITC-dextran (4 kDa; Sigma) assay. All chickens were fasted for 6 h and were administered FITC-dextran (7 mg/kg body weight) by oral gavage 2 h before blood collection. Serum was diluted with phosphate-buffered saline and the concentration of FITC was determined by a fluorescence spectrophotometer with an excitation wavelength of 493 nm and an emission wavelength of 525 nm.

## Thickness measurements of the ileal mucus layer

Thickness of the ileal mucus layer was measured as previously described (6). In brief, the ileum was gently separated and immediately preserved in Carnoy's fixative. The ileum was fixed in Carnoy's fixative for 4 h followed by transfer to fresh Carnoy's for 4 h. The ileum then was washed in methanol twice for 2 h each time and stored at 4°C until further use. Alcian blue staining was performed by the following protocol: deparaffinization at 65°C for 4 h, different alcohol concentration gradients (100%, 95%, 90%, 80%, 70%), dehydration for 2 min, Alcian blue solution for 25 min, washing in running tap water for 1 min, dehydration with 95% and absolute alcohol for 4 min, cleanout in xylene, cover with coverslip. For each sample, five intact mucus layer units were selected for measuring the thickness using a light microscope (Olympus Corporation, Tokyo, Japan) coupled with image processing software (Image J 1.53).

## Histomorphological analysis

For the number of ileal mucin, the Carnoy's fixed sections were stained with the AB-PAS following the manufacturer's instruction (Wuhan Servicebio Technology Co., Ltd, Wuhan, China). The stained slides were measured with light microscope (Olympus Corporation, Tokyo, Japan). The number of mucin was counted manually in five ileal villus per section.

## Quantitative real-time PCR

The ileum mucosa was collected and total RNA was extracted following TRIzol Reagent protocol (AG21102, AG, Changsha, China). cDNA synthesis was performed on 500 ng RNA using Evo M-MLV Reverse Transcriptase Kit (AG11707, AG, Changsha, China). The mRNA expression was analyzed with a SYBR Green Premix Pro Taq HS qPCR Kit (AG11701, AG, Changsha, China) on the iCycler IQ5 (Bio-Rad, Hercules, CA, USA). Primer sequences used in the study are listed in Table S2. Detailed PCR reaction and calculation methods were set according to reported methods (53).

## 16S rRNA sequence processing and analysis

Total DNA was extracted from ileal contents using the E.Z.N.A. soil DNA Kit (Omega Bio-tek, Norcross, GA, USA) according to manufacturer's protocols. Then, DNA concentration and purification were determined with NanoDrop 2000 (Thermo Scientific, Wilmington, USA), and DNA quality was checked by 1% agarose gel electrophoresis. The

V3–V4 region of the bacteria 16S rRNA gene was amplified using primers 338F (5'-ACT CCTACGGGAGGCAGCAG-3') and 806R (5'-GGACTACHVGGGTWTCTAAT-3'). Sequencing of the PCR amplification products was performed on an Illumina MiSeq platform (Illumina, San Diego, USA) according to the standard protocols. 16S raw sequence data were analyzed with QIIME2 platform. Quality control and denoising were performed using DADA2 with default parameters to generate amplicon sequence variants (ASVs). Diversity metrics were calculated using the core-diversity plugin within QIIME2. The principal coordinate analysis based on Bray-Curtis distance and α-diversity was calculated on the free online platform of Microeco Tech Co., Ltd. (Shenzhen, China, https://bioin-cloud.tech/task-list).

## Bile acid quantification in plasma, liver, and ileum samples

Bile acid profiles were measured using LC-MS as previously described (49). Briefly, 0.2 mL plasma, fermentation, and approximately 60 mg liver were extracted with 0.6 mL methanol:water (1:1) extract. After 15 min of vortex and 15 min of centrifugation at $13,000 \times g$ at 4°C, the supernatant was obtained and mixed with methanol:acetonitrile (2:8). The supernatant was filtered through a 0.22 µm nylon filter. Finally, the samples were sealed by LC-MS (Qtrap 5500, AB SCIEX) analysis. Quantification was made using external standard curves.

## Bile acid metabolism analysis by cecal microbiota culture *in vitro*

Fresh cecal samples were obtained from three healthy male broilers fed NF, Con, Glu, Ara, and RS at 21 days and immediately transferred into an anaerobic chamber. Bacteria are obtained by filtration and centrifugation as described above. The 1 mL bacterial suspension was inoculated in 10 mL of brain heart infusion media supplemented with 0.4% of bile juice collected from gallbladders of NF broilers. The *in vitro* cultures were incubated in anaerobic conditions at 37°C for 24 h. After 24 h, the samples were collected and bile acid was analyzed by LC-MS as described above.

## Statistical analysis

Statistical analysis was performed using GraphPad Prism V.8. One-way analysis of variance (ANOVA) with Tukey's analysis was used for parametric ANOVA between groups. The Kruskal-Wallis with Benjamini and Hochberg method was used for non-parametric ANOVA between groups. Data are presented as mean ± SEM. A probability value of $P < 0.05$ or $P < 0.01$ was considered to be statistically significant, which is indicated as follows: *$P < 0.05$, **$P < 0.01$.

## ACKNOWLEDGMENTS

The authors thank the Innovative Research Team of Animal Nutrition & Healthy Feeding (Northwest A&F University, Shaanxi, China) for their assistance in sample collection. The authors thank Microeco Tech Co., Ltd., Shenzhen, China, for sequencing service and assistance. The authors would also like to thank Liru Jian from the State Key Laboratory of Crop Stress Biology for Arid Areas, NWAFU, for LC-MS analyses.

This study was supported by the National Natural Science Foundation of China (31972529 and 32272916) and the Shaanxi Feed Engineering Technology Research Center (2019HBGC-16).

J.Y.: investigation, project administration, formal analysis, data curation, writing-original draft, and visualization. K.Q.: investigation. Y.S.: investigation. X.Y.: conceptualization, project administration, and funding acquisition.

## AUTHOR AFFILIATION

[1]College of Animal Science and Technology, Northwest A&F University, Yangling, Shaanxi, China

## AUTHOR ORCIDs

Jiantao Yang  http://orcid.org/0000-0002-5796-5191

## FUNDING

| Funder | Grant(s) | Author(s) |
|---|---|---|
| MOST | National Natural Science Foundation of China (NSFC) | 31972529, 31172916 | Xiaojun Yang |
| The Shaanxi Feed Engineering Technology Research Center | 2019HBGC-16 | Xiaojun Yang |

## AUTHOR CONTRIBUTIONS

Jiantao Yang, Data curation, Formal analysis, Investigation, Project administration, Visualization, Writing – original draft | Kailong Qin, Investigation | Yanpeng Sun, Investigation | Xiaojun Yang, Conceptualization, Funding acquisition, Project administration

## DATA AVAILABILITY

The 16S rRNA sequences of *in vitro* microbial fermentation and cecal contents in this study were submitted to NCBI Sequence Read Archive under BioProject IDs PRJNA1019452 and PRJNA1019453, respectively.

## ETHICS APPROVAL

All animal protocols were approved by the Animal Care and Use Committee of the College of Animal Science and Technology of the Northwest A&F University (Shaanxi, China).

## ADDITIONAL FILES

The following material is available online.

### Supplemental Material

**Supplemental material (Spectrum02065-23-s0001.pdf).** Tables S1 and S2; Fig. S1 to S3.

### Open Peer Review

**PEER REVIEW HISTORY (review-history.pdf).** An accounting of the reviewer comments and feedback.

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
