## [Reviewer comments · Microbiology Spectrum]

Microbiology Spectrum

Microbiota-accessible fiber activates short chain fatty acid and bile acid metabolism to improve intestinal mucus barrier in broiler chickens

Jiantao Yang, Kailong Qin, Yanpeng Sun, and Xiaojun Yang

Corresponding Author(s): Xiaojun Yang, Northwest A&F University

Review Timeline:

Submission Date:	May 16, 2023
Editorial Decision:	July 31, 2023
Revision Received:	September 22, 2023
Accepted:	November 21, 2023

Editor: Jinxin Liu

Reviewer(s): The reviewers have opted to remain anonymous.

Transaction Report:

DOI: <https://doi.org/10.1128/spectrum.02065-23>

July 31, 2023

Dr. Xiaojun Yang
Northwest A&F University
Yangling
China

Re: Spectrum02065-23 (**Microbiota-accessible fiber activates short chain fatty acid and bile acid metabolism to improve intestinal mucus barrier in broiler chickens**)

Dear Dr. Xiaojun Yang:

Link Not Available

Sincerely,

Jinxin Liu

Journals Department
Reviewer comments:

Reviewer #1 (Comments for the Author):

The manuscript of Yang et al. investigates the role of β -glucan, arabinoxylan, and resistant starch in modulating the gut microbiota and its metabolites to protect the intestinal mucus barrier. The reviewer found several issues that need to be addressed by the authors prior to the publication.

The manuscript is poorly written, and the English language needs to be revised thoroughly. The text contains numerous grammatical mistakes and awkward phrasing, which makes it difficult to understand and follow. Therefore, I recommend that the authors seek assistance from a professional English language editor.

The introduction and discussion need to mention why improving the intestinal mucosal barrier in broiler chickens is important. Preserving chicken gut health, especially after the withdrawal of antibiotic growth promoters from poultry production, is a hot topic for the poultry industry and human consumers. There is a growing body of research investigating strategies to improve poultry gut health by strengthening the intestinal epithelial barrier, modulating the abnormal immune response and inflammation caused by diet and feed additives, and limiting the carriage of food-borne pathogens. Therefore, this manuscript needs to mention the importance of the study for the poultry production and food safety.

The description of the methodology is confusing overall. On line 321, "fresh fecal samples were obtained.."; did the author collect cecum content? Or the whole unit content plus tissue? This point needs to be clarified. The description of the bird study in line 334 needs to be rewritten clearly. A schematic with the experimental design, treatments, and time point of tissues collection would be helpful for the reader to understand the study.

Was the permeability assay with FITC conducted in the same animals used for sample collections? Does the FITC interfere with the analysis conducted in such samples? Two hours of FITC seem to be a short time to evaluate changes in gastrointestinal permeability.

The images in Figure 3A needs to show what the authors measured as the "mucus layer." The scale bars in all the images are very difficult to read.

Staff Comments:

Preparing Revision Guidelines

Please return the manuscript within 60 days; if you cannot complete the modification within this time period, please contact me. If you do not wish to modify the manuscript and prefer to submit it to another journal, please notify me of your decision immediately so that the manuscript may be formally withdrawn from consideration by Microbiology Spectrum.

Title: Microbiota-accessible fiber activates short chain fatty acid and bile acid metabolism to improve intestinal mucus barrier in broiler chickens

The manuscript of Yang et al. investigates the role of β -glucan, arabinoxylan, and resistant starch in modulating the gut microbiota and its metabolites to protect the intestinal mucus barrier. The reviewer found several issues that need to be addressed by the authors prior to the publication.

The manuscript is poorly written, and the English language needs to be revised thoroughly. The text contains numerous grammatical mistakes and awkward phrasing, which makes it difficult to understand and follow. Therefore, I recommend that the authors seek assistance from a professional English language editor.

The introduction and discussion need to mention why improving the intestinal mucosal barrier in broiler chickens is important. Preserving chicken gut health, especially after the withdrawal of antibiotic growth promoters from poultry production, is a hot topic for the poultry industry and human consumers. There is a growing body of research investigating strategies to improve poultry gut health by strengthening the intestinal epithelial barrier, modulating the abnormal immune response and inflammation caused by diet and feed additives, and limiting the carriage of food-borne pathogens. Therefore, this manuscript needs to mention the importance of the study for the poultry production and food safety.

The description of the methodology is confusing overall. On line 321, “fresh fecal samples were obtained.”; did the author collect cecum content? Or the whole unit content plus tissue? This point needs to be clarified. The description of the bird study in line 334 needs to be rewritten clearly. A schematic with the experimental design, treatments, and time point of tissues collection would be helpful for the reader to understand the study.

Was the permeability assay with FITC conducted in the same animals used for sample collections? Does the FITC interfere with the analysis conducted in such samples? Two hours of FITC seem to be a short time to evaluate changes in gastrointestinal permeability.

The images in Figure 3A needs to show what the authors measured as the “mucus layer.” The scale bars in all the images are very difficult to read.

Dear editors and reviewers:

Thanks a lot for your affirmation of our study and kindly comments. We carefully consider the your suggestions and try our best to improve the revised manuscript. Please allow us to respond these questions separately as following.

Reviewer #1 (Comments for the Author):

1. The manuscript of Yang et al. investigates the role of β -glucan, arabinoxylan, and resistant starch in modulating the gut microbiota and its metabolites to protect the intestinal mucus barrier. The reviewer found several issues that need to be addressed by the authors prior to the publication.

Answer: Thanks for your professional opinions and pertinent evaluations. We have answered theses questions and revised the manuscript point by point as you required.

2. The manuscript is poorly written, and the English language needs to be revised thoroughly. The text contains numerous grammatical mistakes and awkward phrasing, which makes it difficult to understand and follow. Therefore, I recommend that the authors seek assistance from a professional English language editor.

Answer: Thanks for your advice. With the help of a professional English language editor, we have revised the language of the manuscript and eliminated possible grammatical or spelling errors.

3. The introduction and discussion need to mention why improving the intestinal mucosal barrier in broiler chickens is important. Preserving chicken gut health, especially after the withdrawal of antibiotic growth promoters from poultry production, is a hot topic for the poultry industry and human consumers. There is a growing body of research investigating strategies to improve poultry gut health by strengthening the intestinal epithelial barrier, modulating the abnormal immune response and inflammation caused by diet and feed additives, and limiting the carriage of food-borne pathogens. Therefore, this manuscript needs to mention the importance of the study for the poultry production and food safety.

Answer: Thanks for your advice. We have added 'the importance of the study for the

poultry production and food safety' to introduction section.

4. The description of the methodology is confusing overall. On line 321, "fresh fecal samples were obtained.."; did the author collect cecum content? Or the whole unit content plus tissue? This point needs to be clarified. The description of the bird study in line 334 needs to be rewritten clearly. A schematic with the experimental design, treatments, and time point of tissues collection would be helpful for the reader to understand the study.

Answer: Thank you for your reminding. In this study, we collected cecal content and revised it on the manuscript. In addition, we have rewritten the description of the bird study and made the schematic as your required (Fig 3A).

5. Was the permeability assay with FITC conducted in the same animals used for sample collections? Does the FITC interfere with the analysis conducted in such samples? Two hours of FITC seem to be a short time to evaluate changes in gastrointestinal permeability.

Answer: The samples we analyzed for FITC came from the same animals as the other indicators. In this study, FITC was not associated with the analysis conducted of other samples, and the same animals were referred to the previous study. (You et al., 2022; Wu et al., 2018). Unlike other animals, poultry have a relatively short digestive tract and food passes through quickly, and the two hours were referred to the previous study (FITC).

You, H. J., Si, J., Kim, J., et al. (2022) *Bacteroides vulgatus* SNUG 40005 restores *Akkermansia* depletion by metabolite modulation. *Gastroenterology*. doi: <https://doi.org/10.1053/j.gastro.2022.09.040>.

Wu, T. R., Lin, C. S., Chang, C. J., et al. (2018) Gut commensal *Parabacteroides goldsteinii* plays a predominant role in the anti-obesity effects of polysaccharides isolated from *Hirsutiella sinensis*. *Gut*. doi: <https://doi.org/10.1136/gutjnl-2017-315458>.

6. The images in Figure 3A needs to show what the authors measured as the "mucus layer." The scale bars in all the images are very difficult to read.

Answer: Thanks for your advice. As shown in Figure 3A, we have marked the measured mucus layers with arrows as your required, and re-marked the scales bars in all the images.

Re: Spectrum02065-23R1 (**Microbiota-accessible fiber activates short chain fatty acid and bile acid metabolism to improve intestinal mucus barrier in broiler chickens**)

Dear Dr. Xiaojun Yang:

Your manuscript has been accepted, and I am forwarding it to the ASM production staff for publication. Your paper will first be checked to make sure all elements meet the technical requirements. ASM staff will contact you if anything needs to be revised before copyediting and production can begin. Otherwise, you will be notified when your proofs are ready to be viewed.

Sincerely,
Jinxin Liu
Editor
Microbiology Spectrum